# Hsp90 Inhibition: A Promising Therapeutic Approach for ARSACS

**DOI:** 10.3390/ijms222111722

**Published:** 2021-10-29

**Authors:** Suran Nethisinghe, Rosella Abeti, Maheswaran Kesavan, W. Christian Wigley, Paola Giunti

**Affiliations:** 1Ataxia Centre, Department of Clinical and Movement Neurosciences, UCL Queen Square Institute of Neurology, Queen Square, London WC1N 3BG, UK; S.Nethisinghe@ucl.ac.uk (S.N.); r.abeti@ucl.ac.uk (R.A.); maheswaran.kesavan.18@alumni.ucl.ac.uk (M.K.); 2Reata Pharmaceuticals Inc., 2801 Gateway Drive, Irving, TX 75063, USA; chris.wigley@reatapharma.com

**Keywords:** ARSACS, ataxia, vimentin, KU-32, Hsp90 inhibition

## Abstract

Autosomal recessive spastic ataxia of Charlevoix-Saguenay (ARSACS) is a neurodegenerative disease caused by mutations in the *SACS* gene, encoding the 520 kDa modular protein sacsin, which comprises multiple functional sequence domains that suggest a role either as a scaffold in protein folding or in proteostasis. Cells from patients with ARSACS display a distinct phenotype including altered organisation of the intermediate filament cytoskeleton and a hyperfused mitochondrial network where mitochondrial respiration is compromised. Here, we used vimentin bundling as a biomarker of sacsin function to test the therapeutic potential of Hsp90 inhibition with the C-terminal-domain-targeted compound KU-32, which has demonstrated mitochondrial activity. This study shows that ARSACS patient cells have significantly increased vimentin bundling compared to control, and this was also present in ARSACS carriers despite them being asymptomatic. We found that KU-32 treatment significantly reduced vimentin bundling in carrier and patient cells. We also found that cells from patients with ARSACS were unable to maintain mitochondrial membrane potential upon challenge with mitotoxins, and that the electron transport chain function was restored upon KU-32 treatment. Our preliminary findings presented here suggest that targeting the heat-shock response by Hsp90 inhibition alleviates vimentin bundling and may represent a promising area for the development of therapeutics for ARSACS.

## 1. Introduction

Autosomal recessive spastic ataxia of Charlevoix-Saguenay (ARSACS; OMIM 270550) is an inherited neurodegenerative disorder characterised by early-onset cerebellar ataxia with spasticity, peripheral neuropathy, dysarthria and nystagmus [1]. ARSACS is caused by mutations in the *SACS* gene, which encodes sacsin, a 520 kDa modular protein containing, from the N- to C-terminus, a ubiquitin-like (UbL) domain that binds to the proteasome [2], three large sacsin repeat regions (SRRs) that may have an Hsp90-like chaperone function [3], a J-domain that binds Hsp70 [2,4], and a higher eukaryotes and prokaryotes nucleotide-binding (HEPN) domain that can dimerise [2,3,4,5,6]. Although the function of sacsin is still to be elucidated, the presence of these conserved domains, often found in molecular chaperones and components of the ubiquitin–proteasome pathway, suggest a role in proteostasis. Cytoskeletal abnormalities have been previously reported in mouse models of ARSACS [7,8] and patient dermal fibroblasts [9]. The striking bundling and collapse of the vimentin intermediate filament network observed in patient dermal fibroblasts mirrors the abnormal bundling of neurofilaments in multiple neuronal populations in *Sacs* knockout mice and patients with ARSACS [7]. Vimentin is a type III intermediate filament that is found in mesenchymal, endothelial and haematopoietic cell types, with a role in supporting cellular membranes, anchoring some organelles in the cytoplasm, and transmitting membrane receptor signals to the nucleus [10]. Neurofilaments are, however, type IV intermediate filaments found in neuronal axons [10]. A recent study has proposed that there are distinct, but complementary, functions of sacsin domains in intermediate filament organisation and turnover [8]. That study also proposed the upregulation of heat-shock protein chaperones could compensate for the loss of sacsin function and restore the intermediate filament network [8].

Heat-shock protein 90 (Hsp90) is the master regulator of the heat-shock response (HSR) since it binds heat-shock factor 1 (HSF1), maintaining it in an inactive form. Cellular stress disrupts the Hsp90–HSF1 complex, leading to the trimerisation and activation of HSF1 and upregulation of molecular chaperone gene expression, including Hsp70, a signature of the HSR [11]. Small-molecule N-terminal Hsp90 inhibitors can mimic cell stress and promote the release of HSF1 from Hsp90 to activate the HSR [12]. The induction of molecular chaperones by the HSR can reduce protein aggregation; hence, Hsp90 inhibitors have been considered potential therapeutics for neurodegenerative diseases where the accumulation of specific misfolded or aggregated proteins is a key feature [13,14,15,16]. KU-32 is a novobiocin-derived C-terminal-targeted Hsp90 inhibitor that can potently induce Hsp70 and protects against neuronal cell death [17,18,19].

Here, we used vimentin bundling as a biomarker for testing the therapeutic potential of Hsp90 inhibition by KU-32. We found that treatment with KU-32 significantly reduces the vimentin bundling phenotype observed in ARSACS patient fibroblasts. We also found that mitochondrial membrane potential maintenance was compromised in ARSACS patient fibroblasts but could be rescued upon treatment with KU-32. Our preliminary findings suggest that targeting the HSR by C-terminal-targeted Hsp90 inhibition is a promising area for further exploration and therapeutic development.

## 2. Results

### Effect of KU-32 Treatment

To assess the therapeutic potential of C-terminal Hsp90 inhibition, dermal fibroblast lines from a control, a carrier, and two patients were cultured for 24 h with either DMSO (Vehicle) or 100 nM KU-32 prior to the immunofluorescence detection of vimentin intermediate filaments by confocal imaging. This concentration of KU-32 was able to induce Hsp70, and a mild increase in sacsin expression was also observed (Appendix A). Representative images are shown in Figure 1A with examples of cells displaying a disorganised vimentin network (bundling) indicated with arrows. The quantification of cells with the bundled vimentin phenotype is shown in Figure 1B. Vimentin bundling was significantly higher in all the cell lines relative to the control cell line. Notably, this included the carrier cell line. Interestingly, KU-32 treatment significantly reduced the levels in the carrier and patient cell lines but did not alter the vimentin bundling levels in the control cell line (Figure 1B).

We then measured the mitochondrial membrane potential (ΔΨ_m_) maintenance to understand whether the alterations in vimentin were associated with mitochondrial dysfunction. Using the fluorescent probe tetramethylrhodamine methyl ester (TMRM), we assessed the maintenance of the ΔΨ_m_ in Control and ARSACS Patient fibroblasts (Figure 2A). TMRM is a permeable, cationic dye sequestered by active mitochondria, which redistributes into the cytosol during depolarisation, decreasing the fluorescent signal in the mitochondria. After baseline measurements, we challenged the cells with oligomycin (an ATP-synthase inhibitor) to test the electron transport chain (ETC)’s ability to maintain the ΔΨ_m_. ARSACS patient fibroblasts showed a decrease in TMRM intensity following oligomycin administration (Figure 2A). This revealed the inability of the ETC to maintain the ΔΨ_m_ in ARSACS patient cells; however, this was restored by pre-incubation with KU-32. The histogram in Figure 2B shows the percentage of the increased depolarisation upon oligomycin administration. This was calculated as the difference in TMRM fluorescence intensity after oligomycin administration from the baseline. Although the patient fibroblasts showed clear mitochondrial dysfunction with an increased oligomycin response (Patient 1, 223 ± 71; Patient 2, 270 ± 93%; mean ± standard deviation), KU-32 treatment was able to prevent oligomycin-induced depolarisation (Patient 1, 103 ± 14; Patient 2, 123 ± 24%), indicating an improvement in mitochondrial function (Figure 2B). Carrier showed a similar pattern to the patients, but there was no significant difference compared to control (Carrier, 171 ± 47; Carrier KU-32, 91 ± 32).

To corroborate our results, we measured the mitochondrial mass by looking at the mitochondrial and the total volume cell by cell. Figure 3A shows the mitochondrial network compared to control, perhaps suggesting an impairment in mitochondrial quality control. Moreover, the histogram shows a significant increase in mitochondrial volume in Patient 1, confirming the mitochondrial defect (Figure 3B). However, KU-32 successfully restored the volume to control levels. Although the carrier line shows a similar trend to the patient line, it is not significant compared to the control.

## 3. Discussion

It has previously been observed that ARSACS patient dermal fibroblasts display a collapsed, disorganised vimentin intermediate filament network [9], like the abnormal neurofilament bundling found in neurons from *Sacs*-knockout mice and ARSACS patients [7]. Here, we used the striking phenotype of vimentin bundling as a biomarker of the disease state to assess the efficacy of a C-terminal-targeted Hsp90 inhibitor, KU-32, in treating ARSACS. We found that KU-32 treatment significantly reduced vimentin bundling in both carrier and patient cell lines. This suggests that targeting Hsp90 may represent a promising avenue for therapeutics in ARSACS and that the assay confirms that vimentin bundling is a very good biomarker of ARSACS. Interestingly, the carrier cell line under vehicle-treatment conditions had significantly more vimentin bundling than control cells, although much less than that seen in patients, despite ARSACS being an autosomal recessive disorder and carriers being asymptomatic. We have previously shown that healthy carriers may show intermediate phenotypes, with mild thickening of the retinal nerve fibre layer (RNFL), as determined by optical coherence tomography (OCT) in two carriers [20]. However, the further examination of 13 heterozygous carriers revealed that this thickness (115.5 µm) was just below the 119 µm threshold for distinguishing ARSACS from non-ARSACS patients [21]. A recent study suggests that this cut-off should be increased to 121 µm to provide 100% accuracy in diagnosing ARSACS [22].

The role of sacsin in intermediate filament assembly and dynamics has previously been examined in *Sacs*-knockout mouse motor neurons [8]. The study found that increasing the expression of heat-shock proteins could compensate for sacsin deficiency and resolve neurofilament bundling [8]. In addition to the ectopic expression of HSPA1A, the authors also tested celastrol, an inducer of multiple HSPs, including HSPA1A, DNAJB1 and HSPB1 [23]. Celastrol has also been shown to inhibit Hsp90 through its suppression of the interaction between Hsp90 and its co-chaperone Cdc37 [24,25]. Culturing *Sacs*^−/−^ motor neurons with 1 µM celastrol for 24 h was able to induce the expression of HSPA1A and resolve neurofilament bundles.

In addition to the altered intermediate filament network, ARSACS patient cells also show changes in mitochondrial morphology, dynamics and distribution resulting in impaired oxidative phosphorylation and increased oxidative stress [7,26,27,28]. This may be linked to the cytoskeletal abnormalities seen in ARSACS since the cytoskeleton regulates mitochondrial dynamics [29,30,31]. It is possible that Hsp90 inhibition by KU-32 could also correct mitochondrial dysfunction in ARSACS patient cells [19]. The activity of KU-32 has been shown to be dependent upon the Hsp70 expression [19]. That study hypothesised that an interaction between Hsp70 and Drp1 (dynamin-related protein 1) may aid the refolding of the latter or prevent its aggregation [19]. Mitochondrial fission is an important step in maintaining mitochondrial health and is dependent on the recruitment of Drp1 to potential sites of division [27,32]. Sacsin may play a role in recruiting or retaining Drp1 at mitochondrial fission sites, and its mutation in ARSACS leads to the mitochondrial changes described above [7,26,27,28]. In line with this, our experiments here show that mitochondrial membrane potential maintenance was compromised in ARSACS patient fibroblasts but could be rescued upon treatment with KU-32, illustrating the dual nature of the therapeutic benefit KU-32 offers. Moreover, another indication of mitochondrial defect was achieved by measuring the mitochondrial volume, which was increased in ARSACS patient fibroblasts. Interestingly, KU-32 was able to re-establish the mitochondrial volume to control levels. We believe that mitochondrial membrane potential maintenance together with vimentin bundling is a useful marker for the future screening of novel compounds for treating ARSACS.

Therapies for patients with ARSACS are urgently needed, and this study shows that targeting the HSR, in particular, Hsp90, is a promising avenue. Advances in this domain would pave the way for treatments that will have a significant impact on the lives of patients with ARSACS.

## 4. Materials and Methods

### 4.1. Ethics Statement

The ARSACS patient and control dermal fibroblast lines were collected under ethical approval from the NRES Committee London-Harrow for the European integrated project on spinocerebellar ataxias, EUROSCA (NHS REC reference: 04/Q0505/21). Written informed consent was obtained from the patients for the skin biopsies.

### 4.2. Cell Culture

Fibroblast lines were cultured in Dulbecco’s Minimum Eagle Medium (DMEM) with high glucose and GlutaMAX, supplemented with 15% (*w*/*v*) foetal bovine serum (FBS) and MEM non-essential amino acids (all supplied by Thermo Fisher Scientific, Waltham, MA, USA). All the cells were maintained in a constant humidified atmosphere of 5% CO_2_ at 37 °C. Since ARSACS is a recessive disorder and carriers are healthy, we included a carrier line to act as a benchmark for the ability for the KU-32 therapeutic rescue of the ARSACS phenotype. Additionally, we also included a fibroblast cell line from a healthy control. ARSACS patient lines 1 and 2 are from siblings with the mutations c.8339T>C (p.Phe2780Cys) and c.12416T>C (p.Leu4139Ser) on one allele and c.11675C>G (p.Ser3892X) on the other allele. The ARSACS carrier line was heterozygous for the mutation c.5151dupA (p.Ser1718fsX1736). The sacsin levels in all four fibroblast lines were assessed by immunoblotting (Appendix A). The patient fibroblast lines showed reduced sacsin levels, and the carrier line had an intermediate level of sacsin compared to the control line.

KU-32 was provided by Reata Pharmaceuticals Inc. as a 10 mM stock in DMSO. Cells were treated with either DMSO vehicle (Sigma-Aldrich, Dorset, UK) or KU-32 diluted to 100 nM in fibroblast medium for 24 h prior to fixation or mitochondrial membrane potential experiments. For immunoblot analysis, cells were cultured with DMSO vehicle (Sigma-Aldrich, Dorset, UK), 100 nM KU-32, 500 nM geldanamycin (Sigma-Aldrich, Dorset, UK) or 1 µM celastrol (Sigma-Aldrich, Dorset, UK) diluted in fibroblast medium for 24 h prior to lysis.

### 4.3. Immunofluorescence and Imaging

Cells were cultured on CellCarrier Ultra 96-well cyclic olefin-bottomed microplates (PerkinElmer Inc., Waltham, MA, USA) and were fixed with 4% formaldehyde in PBS for 15 min at room temperature. The wells were washed thrice with PBS, pH 7.4, for 10 min each time with gentle agitation at room temperature. The cells were permeabilised with 0.5% (*v*/*v*) Triton X-100 for 15 min at room temperature with gentle agitation. The wells were washed thrice with PBS, pH 7.4, for 10 min each time with gentle agitation at room temperature. The cells were incubated with blocking buffer (3% (*w*/*v*) BSA, fraction V, in DPBS) for 1 h at room temperature with gentle agitation. The cells were then incubated overnight at 4 °C with gentle agitation with immunofluorescence staining solution containing a 1:1000 dilution of Alexa Fluor 488-conjugated mouse anti-vimentin (BioLegend, San Diego, CA, USA; #677809) and 300 nM DAPI counterstain in blocking buffer. The wells were washed thrice with PBS, pH 7.4, for 10 min each time with gentle agitation at room temperature. A 100 µL volume of PBS, pH 7.4, was added to each well, and the plates were sealed, protected from light and stored at 4 °C until imaged. Confocal images were acquired using an Opera Phenix™ high content confocal microscope (PerkinElmer Inc., Waltham, MA, USA) with a 40× air objective.

### 4.4. Mitochondrial Membrane Potential Assay

The mitochondrial membrane potential (ΔΨ_m_) was measured with tetramethyl rhodamine methyl ester (TMRM, 25 nM; Invitrogen, Waltham, MA, USA) in ‘redistribution mode’ [33]: The dye was allowed to equilibrate for 45 min prior to the experiment and was present continuously in the recording solution. TMRM distributes between cellular compartments in response to different potentials and, at concentrations <50 nM, in healthy cells, the fluorescent signal shows a localisation to the mitochondria, where it is retained until depolarisation induced by mitotoxins. The basal level of ΔΨ_m_ was measured by exciting TMRM at 560 nm and collecting the images with a 590 nm long-pass filter. The maintenance of ΔΨ_m_ was measured after using 2 µg/mL oligomycin, 1 µM rotenone and 1 µM FCCP.

### 4.5. Mitochondrial Volume Assay

Cells were stained with TMRM (25 nM; to visualise mitochondria) and Calcein Blue-AM (5 μM; to visualise the cell body) in cell culture media, respectively, for 45 min and 20 min. Z-stacks images were acquired using a 710 Zeiss Confocal microscope and analysed with the Volocity software (version 6.0, PerkinElmer Inc.). In detail, TMRM was visualised using an excitation wavelength of 561 nm, and Calcein Blue-AM was excited at 405 nm. Analysis was performed for each channel, measuring the volume via the Volocity software, and then, the ratio between the mitochondrial portion and the total cell volume was determined. Five fields of view per dish were analysed. To avoid observer bias, the analysis was blinded. The means represent >3 independent experiments.

### 4.6. Immunoblotting

Cells were lysed in CelLytic M (Sigma-Aldrich, Dorset, UK) supplemented with protease and phosphatase inhibitors (Roche, Mannheim, Germany). A 20 µg amount of lysate was separated on 3–8% gradient NuPAGE Tris-Acetate gels (Invitrogen, Waltham, MA, USA) and transferred to nitrocellulose membranes (LI-COR, Lincoln, NE, USA). Revert 700 Total Protein Stain (LI-COR) was used to assess the transfer and for immunoblot normalisation. The membranes were blocked with Intercept (TBS) Blocking Buffer (LI-COR) before probing with the specified antibodies diluted in Intercept T20 (TBS) antibody Diluent (LI-COR). Primary antibodies were used at the following titres: 1:1500 for rabbit monoclonal anti-sacsin (abcam, Cambridge, UK; #ab181190), 1:5000 for mouse monoclonal anti-GAPDH (Invitrogen, #MA-15738), and 1:1000 for mouse monoclonal anti-Hsp70 (abcam; #ab2787) and rabbit polyclonal anti-pan-AKT (abcam; #ab8805). The secondary antibodies used were goat anti-mouse IRDye 680RD (abcam; #ab216776) and goat anti-rabbit IRDye 800CW (abcam; #ab216773). Images were acquired using an Odyssey CLx Imaging System (LI-COR), and analysis was performed using the Empiria Software (version 2.1.0, LI-COR).

### 4.7. Statistical Analysis

Ten fields of view from 3 wells for each treatment condition were analysed. Vimentin bundling counts were performed by 3 operators (S.N., R.A. and M.K.) in a blinded manner to ensure unbiased analysis. The data were subjected to two-way ANOVA with Tukey’s multiple-comparison post hoc analysis. The oligomycin response was calculated as a percentage, normalised to the vehicle-treated control. An ordinary one-way ANOVA with Dunnett’s multiple-comparison post hoc analysis was performed on the oligomycin response data. The error bars on all the graphs indicate the mean ± standard deviation. Where indicated, * = *p* < 0.05, ** = *p* < 0.01 and **** = *p* < 0.0001. All the statistical analyses were performed using the Prism statistical software (version 9.1.0, GraphPad).

## Figures and Tables

**Figure 1 ijms-22-11722-f001:**
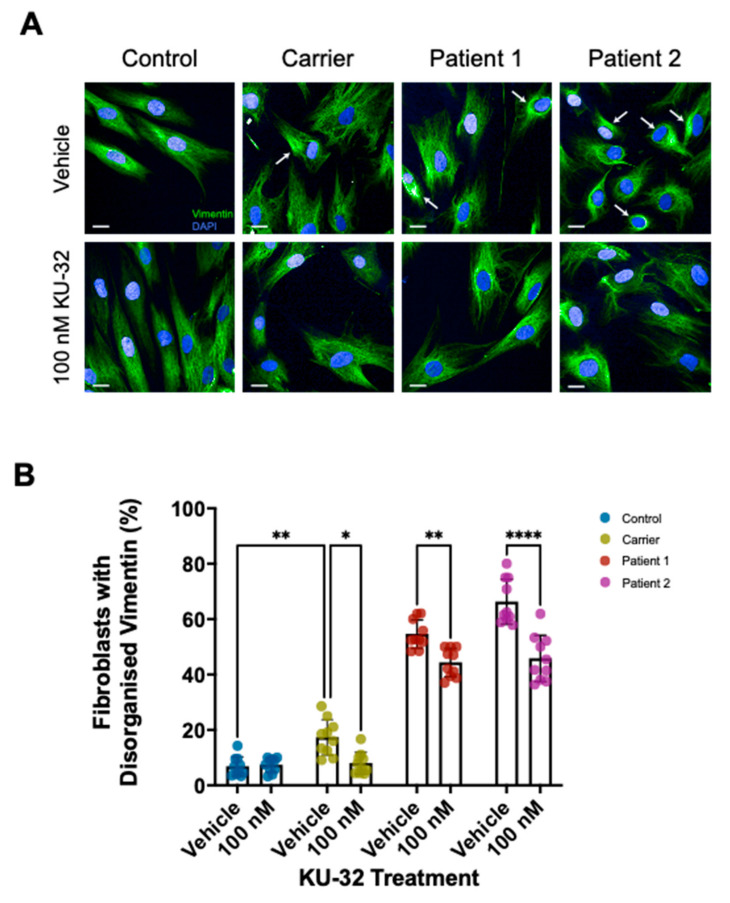
**Reduction of vimentin bundling in dermal fibroblasts by KU-32**. Fibroblast lines were cultured for 24 h with either DMSO vehicle or 100 nM KU-32 prior to fixation and immunofluorescence. (**A**) Representative micrographs of Control, Carrier, Patient 1 and Patient 2 fibroblast lines with and without KU-32 treatment (Vimentin, green; DAPI, blue). Examples of cells with vimentin bundling are indicated with arrows. Scale bar = 20 µm. (**B**) The percentage of cells with an abnormal vimentin network was then quantified for each cell line and treatment condition. Ten fields of view comprising >200 cells per cell line and treatment were analysed. Conditions were compared by two-way ANOVA with Tukey’s multiple-comparison post hoc analysis. KU-32 treatment significantly reduced vimentin bundling in the carrier and both patient lines but did not alter vimentin bundling in the control line. However, the vimentin bundling levels in the treated patient lines remained significantly higher than in control and carrier lines. * *p* ≤ 0.05; ** *p* ≤ 0.01; **** *p* ≤ 0.0001.

**Figure 2 ijms-22-11722-f002:**
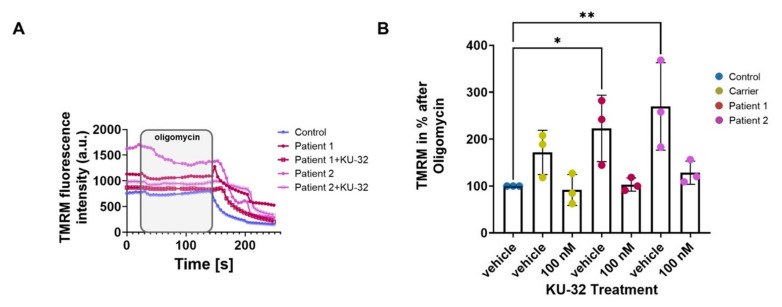
**ARSACS patient fibroblasts show a significantly deeper depolarisation after oligomycin administration, which is significantly reduced upon 100 nM KU-32 treatment.** Fibroblast lines were cultured for 24 h with either DMSO vehicle or 100 nM KU-32 prior to equilibration with 25 nM TMRM for 45 min and imaging. (**A**) Kinetic curves of the TMRM fluorescence intensity of vehicle-treated control, carrier, and ARSACS Patient 1 and 2 fibroblasts following the addition of oligomycin (2 µg/mL), rotenone (5 µM) and FCCP (1 µM). (**B**) Graph showing the increase in depolarisation after oligomycin administration in percentage normalised to control. * *p* ≤ 0.05; ** *p* ≤ 0.01.

**Figure 3 ijms-22-11722-f003:**
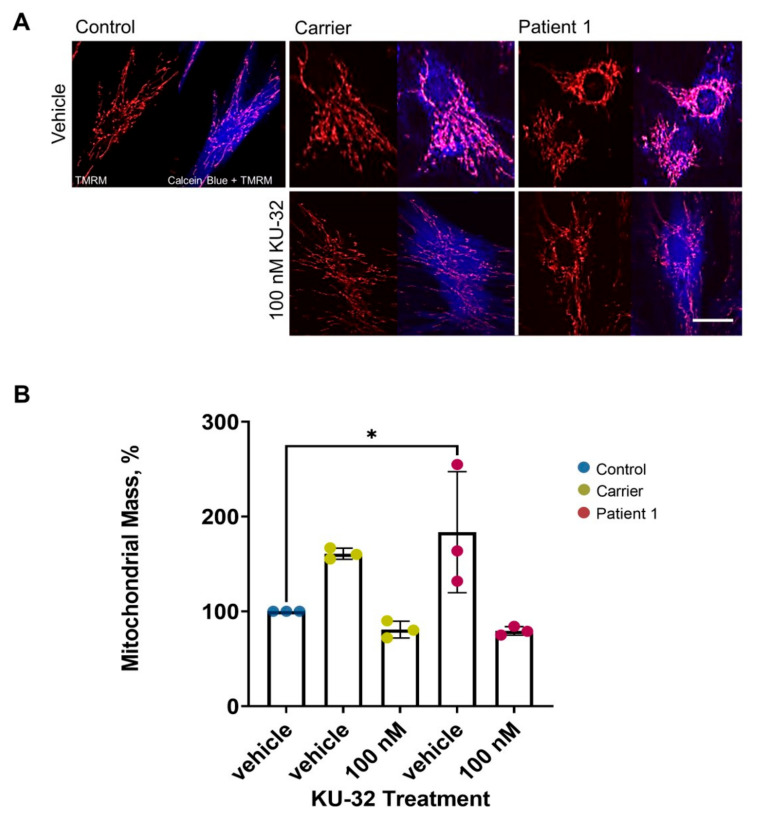
**ARSACS patient fibroblasts show an increased mitochondrial volume compared to control, which was reduced upon 100 nM KU-32 treatment.** (**A**) The micrographs show the mitochondrial staining (TMRM; red) and the double staining with the cell body (Calcein Blue; blue). (**B**) The histogram represents the calculation of the mitochondrial mass compared to the total volume of the cell body as a percentage. (Carrier vehicle, 160 ± 5.8; Carrier 100 nM KU-32, 80.8 ± 9; Patient 1 vehicle, 184 ± 63; and Patient 1 100 nM KU-32, 79.4 ± 4.5). * *p* ≤ 0.05. Scale bar = 20 µm.

## Data Availability

The data presented in this study are available from the corresponding author upon reasonable request.

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
