# Peer review of "Hsp90 Inhibition: A Promising Therapeutic Approach for ARSACS"

_ijms, 2021, doi:10.3390/ijms222111722_

Round 1
Reviewer 1 Report
In this paper, Suran Nethisinghe et al. study the effects of Hsp90 inhibition in ARSACS patients and carrier, identifying two readouts useful for screening compounds for treating ARSACS.
In particular, they show therapeutic benefit of KU-32 compound to rescue both the abnormal vimentin network and the mitochondrial membrane potential maintenance linking the cytoskeletal abnormalities and mitochondrial dysfunction seen in ARSACS. These issues represent two milestones on which are based the recent experimental approaches in ARSACS and the study of common parts and links of this intricate molecular networks is crucial for our understanding of the pathology.
Despite my appreciation of overall intent of the paper, it doesn’t seem to be designed, in particular from a methodological point of view, to demonstrate adequately a promising therapeutic approach. My impression is that this is a very preliminary work of limited use for the ARSACS community.
The paper lacks of a preliminary or supplementary samples characterization. In particular what are sacsin mRNA/protein levels in the analyzed samples? Did the authors obtain modifications in sacsin mRNA/protein levels following the KU-32 treatment? What is the mutation in the two patients? How is their metabolic state (OCR for example)
Literature reports the dose of 100nM of KU-32, as the more appropriate to protect neuronal cells against Aβ-induced toxicity. However, the authors do not verify if KU-32 100nM is able to increase the protein level of HSP70 or other HSPs, in tested fibroblast primary lines. A western blot analysis is strictly recommended to demonstrate that the compound is active in the analyzed system, regardless of the obtained results.
Did they observe a dose response effect (if the effects are specific, one would expect a dose response)? Any titration experiments should be performed also using a similar antibiotic with no effects on HSR in parallel, discussing about the specificity of the used compound.
Moreover, they should expand the study on the mitochondrial function, maybe including an evaluation of mitochondrial network or a micro-​oximetry analysis (also in presence of KU-32). These tests, better than the measure of mitochondrial membrane potential alone, could provide a more complete data on mitochondrial function and the possible rescue by treatments with active compounds.
In figure 1, authors performed an immunofluorescence detection of vimentin in presence or absence of KU-32 treatment. They also show a histogram reporting the changes in the percentage of fibroblasts with disorganizes vimentin after treatment. In this case, they should propose an unbiased method to distinguish normal cells and cells with an abnormal vimentin network. In spite of the numerousness of the analyzed fields, it seems to appear as a visual or quick result. Please explain in the methods section
ARSACS is a recessive condition. How do the authors explain the apparent increase of vimentin bundling in a carrier individual? What do they imply by stating that carriers show intermediate phenotypes?
Although they well discuss the possibility to identify in carriers, intermediate phenotypes, they should better explain why they want to promote vimentin bundling from an interesting biomarker in ARSACS (as suggested by literature), to an appropriate readout for testing a potential therapeutic role of HSP-90 inhibition. As far as I know they are healthy…
Figure 2 reports a double experiment (kinetics and end-point) to assess the mitochondrial membrane potential in patient 1. They should report in 2A the tracks related to patient 2, carrier and all the tracks obtained after KU-32 treatment and quantify the cell depolarization in 2B.
However, it is unclear what method was used to quantify depolarization after oligomycin in figure 2B. Please provide it together with the unit of measure of y-axis.
Reviewer 2 Report
The manuscript "Hsp90 inhibition: a promising therapeutic approach for 2
ARSACS" is an interesting communication that investigates the beneficial effects of HSP90 nhibition as a therapeutical interventin for Charlevoix-Saguenay ataxia (ARSACS). The manuscript however presents limited data and thus the discussion appears to be speculative:
- A viability assay for KU-35 in fibroblast should be addresed.
- Figure 2B does not show any significance. However, * is decribed in the figure legends and results are described as significant. Please clarify and adjust the text accordingly.
- Investigation in one pathway of HSP90 (HSP70, active rhoA, ERK or AKT) should be performed to justify the therapeutical intervention of KU-35 in fibroblast (https://pubmed.ncbi.nlm.nih.gov/24263156/ , https://www.mdpi.com/2073-4409/10/6/1489 )
Author Response
Please see the attachement.

Round 2
Reviewer 1 Report
It would be valuable to add a supplementary figure including some representarive images of the mitochondrial volume assey detected in control, carrier and patient(s) w and w/o KU-32. This could provide further information about shape and distribution of mt network too.
Author Response
We thank the reviewer for their suggestion. We have now added the representative images to Figure 3. They now show the increase in mitochondrial volume in Carrier and Patient 1, and that the mitochondrial network is more condensed than Control. Perhaps, this might be due to the inhibition of mitochondrial quality control.
Reviewer 2 Report
No more comments.
Author Response
We thank the Reviewer for their time assessing our manuscript and are pleased that they have no further comments to address.